# Single–Molecule Study of DNAzyme Reveals Its Intrinsic Conformational Dynamics

**DOI:** 10.3390/ijms24021212

**Published:** 2023-01-07

**Authors:** Yiming Zhang, Zongzhou Ji, Xin Wang, Yi Cao, Hai Pan

**Affiliations:** 1Wenzhou Institute, University of Chinese Academy of Sciences, Wenzhou 325001, China; 2Wenzhou–Kean University, Wenzhou 325060, China; 3Northeastern University, Shenyang 110819, China; 4Jinan Microecological Biomedicine Shandong Laboratory, Shounuo City Light West Block, Qingdao Road 3716#, Huaiyin District, Jinan 250117, China; 5National Laboratory of Solid–State Microstructure, Department of Physics, Nanjing University, Nanjing 210093, China

**Keywords:** single–molecule fluorescence resonance energy transfer, DNAzyme, conformational dynamics, metal ions

## Abstract

DNAzyme is a class of DNA molecules that can perform catalytic functions with high selectivity towards specific metal ions. Due to its potential applications for biosensors and medical therapeutics, DNAzyme has been extensively studied to characterize the relationships between its biochemical properties and functions. Similar to protein enzymes and ribozymes, DNAzymes have been found to undergo conformational changes in a metal–ion–dependent manner for catalysis. Despite the important role the conformation plays in the catalysis process, such structural and dynamic information might not be revealed by conventional approaches. Here, by using the single–molecule fluorescence resonance energy transfer (smFRET) technique, we were able to investigate the detailed conformational dynamics of a uranyl–specific DNAzyme 39E. We observed conformation switches of 39E to a folded state with the addition of Mg^2+^ and to an extended state with the addition of UO_2_^2+^. Furthermore, 39E can switch to a more compact configuration with or without divalent metal ions. Our findings reveal that 39E can undergo conformational changes spontaneously between different configurations.

## 1. Introduction

Deoxyribozymes, also known as DNAzymes, DNA enzymes, or catalytic DNA, have been established as biological catalysts acting similarly to other protein enzymes and ribozymes [1]. Their sequences are selected through in vitro selection methods [2,3,4,5]. Because of their stability in physiological conditions, high sensitivity, and selectivity, DNAzymes are widely applied in practices [6], such as heavy metal ion detectors [7,8], biosensors [9,10,11], cellular imaging [12,13,14] and diagnostics [15,16].

As a cofactor, metal ions are critical for ribozyme and DNAzyme activities. Both previous bulk and single–molecule studies of DNAzyme have tried to better understand the significance of metal ions in catalysis [17,18,19,20,21,22,23,24]. Studies of 8–17 DNAzyme discussed two different pathways of 8–17 DNAzyme to activate in the presence of Zn^2+^ (or Mg^2+^) and Pb^2+^ [17,22]. In particular, the specific metal–ion–dependent activities of 8–17 DNAzyme can be listed in the following orders: Pb^2+^ >> Zn^2+^ >> Mg^2+^. With the presence of Zn^2+^ or Mg^2+^, 8–17 DNAzyme undergoes a global folding conformation, whereas no folding occurs in the presence of Pb^2+^ [22]. This might be due to the prearrangement of 8–17 DNAzyme for the quick binding of Pb^2+^ without any conformational changes needed to activate the cleavage function.

A recent study introduced another in vitro selected DNAzyme, 39E, specific to UO_2_^2+^ ions [25,26]. Like other DNAzymes, 39E is composed of an enzyme strand and a substrate strand forming a three–way junction formation (Figure 1A). The two arm stems (stem I and III) are constructed by the enzyme strand and the substrate strand through complementary pairing. The additional 30 nucleotides in the enzyme strand form stem II, containing a catalytic core in the bulge loop. Fluorescence studies reported that 39E has a detection limit down to 45pM with remarkably high selectivity (only to Th^2+^ and UO_2_^2+^) [25]. Bulk fluorescence resonance energy transfer (FRET) experiments also investigated the metal–ion–dependent global folding of 39E in the presence of different metal ions [27]. Although 39E has no cleavage activities with Mg^2+^ or Zn^2+^, it showed a folding behavior of stem I and stem II while Mg^2+^ or Zn^2+^ was present. However, no folding between stem I and stem II was observed with UO_2_^2+^ alone, which is similar to the case of 8–17 DNAzyme. In contrast, stem I and III underwent a modest folding with UO_2_^2+^ presence. Researchers have additionally demonstrated that, with a lower Na^+^ concentration, UO_2_^2+^ or Pb^2+^ could prompt a slight unfolding of 39E. This indicates that with different ionic strengths, 39E may go through different activation pathways. However, the structural configurations of DNAzymes may not be optimized since they are products of in vitro selections. There might be several local energy minimums that can trap the molecules into a specific configuration. A recent study revealed the crystal structure of DNAzyme 8–17 and proposed three catalytic forms of 8–17 DNAzyme [28]. Another study showed that the insertion of a C3 spacer in the catalytic core did not corrupt 39E activity but enhanced about 10–fold UO_2_^2+^–sensitivity [29]. Some simulation works also investigated the effects of pH values and monovalent metal ions on the activities of DNAzymes [26,27,30,31]. These suggest that DNAzymes might adopt much more complicated configuration arrangements toward the catalytic functions. Nevertheless, such crucial information might not be distinctly characterized through bulk experiments considering the limited accessibilities. Over the last two decades, single–molecule techniques have become very powerful methods for studying protein folding/unfolding [32,33], intra/inter–molecular interactions [34], and biomolecule conformational changes [35,36]. It can be used to probe much more detailed structural information than conventional bulk methods, which could provide insightful inspirations for the design of novel biomaterials [37].

Herein, we applied the single–molecule FRET (smFRET) technique [38] to further probe the conformational changes of 39E. The bulk experiments have reported that there were not many FRET signal changes between stem I and stem III. Thus, we labeled the enzyme strand of 39E with biotin at the 3′ end, an acceptor (Cy5) at the middle of stem II, and the substrate strand of 39E with a donor (Cy3) at the 5′ end in stem I (Figure 1A). Through this labeling strategy, we were able to monitor the distance changes between stem I and stem II. Since the chemical properties and the overall UO_2_^2+^–dependent cleavage activity of 39E have been characterized well in the previous reports, we were only interested in the conformational changes of 39E in this study. Therefore, we replaced the cleavage site ribonucleotide adenine (rA) with a deoxynucleotide adenine (A) to keep the substrates intact during the process. We observed a global folding of DNAzyme 39E by adding Mg^2+^ ion in the solution and detected an extending behavior in the presence of UO_2_^2+^, which is consistent with the results from previous bulk experiments. Intriguingly, 39E can undergo conformational changes spontaneously to a more compact state or an extended state with or without divalent metal ions. Our results revealed the intrinsic conformational dynamics of 39E.

## 2. Results and Discussion

### 2.1. Metal–Ion–Dependent Folding Induced by Mg^2+^ and Extending by UO_2_^2+^

Metal ions are involved in many critical biological processes, such as signal transduction and enzymatic activity. It is been found that metal–ions–dependent conformational changes play a critical role in the activities of proteins, DNA (DNAzymes [26,27] and Holliday junctions [39,40]), and RNA (ribozymes [41] and riboswitches [42]). The in vitro selected DNAzyme 39E has been isolated to have specificity to UO_2_^2+^. A previous report showed its global folding behavior upon interaction with Mg^2+^ [27]. To obtain a deeper insight into the 39E working mechanism, we applied smFRET to investigate the detailed conformational changes induced by metal ions. The DNA construct was labeled with donor fluorophore Cy3 at the 3′ end of the substrate strand and acceptor fluorophore Cy5 in the middle of the enzyme strand (Figure 1A). The annealed 39E DNA molecules were immobilized on the biotin–PEGlyated glass surface through streptavidin–biotin interaction. The reaction was triggered by a flow containing 30 mM Mg^2+^ in the imaging buffer using a syringe pump 13 s after the recording started. Most traces (62%, n = 752 out of 1213) were similar to the one represented in Figure 1B. Before injecting Mg^2+^ in the flow cell, 39E was in a native state (N) with FRET efficiency (E_FRET_) of 0.25. After adding 30 mM Mg^2+^, E_FRET_ increased to 0.47 immediately (Figure 1D), indicating an intact folded formation (F) of 39E in which the donor and acceptor became closer, which is consistent with the bulk FRET measurements.

Since 39E is a uranyl–specific DNAzyme, we also applied 10 μM UO_2_^2+^ to the reaction buffer. Interestingly, although non–cleavable substrates were used, we were still able to observe traces (47%, n = 286 out of 609) with decreased E_FRET_ (from 0.24 to 0.11) presenting an extended state (E) of 39E upon binding UO_2_^2+^ (Figure 1C,E). This is different from Pb^2+^–specific DNAzyme 8–17, of which Pb^2+^ has no effect on E_FRET_ with non–cleavable substrates. Among the traces showing extension induced by UO_2_^2+^, 46% of them (n = 131 out of 286) exhibited some delays of change in E_FRET_ after adding UO_2_^2+^ (Appendix A). Compared to 8–17 DNAzyme, a shorter lifetime of the delay was observed, suggesting a faster conformational rearrangement of 39E to the extended state.

### 2.2. Conformational Switches with Mg^2+^ Presence

As mentioned above, 62% of the traces showed folding activities of 39E with 30 mM Mg^2+^ in the solution. Unexpectedly, the remaining 38% of the traces (461 out of 1213) showed conformation dynamics of 39E when Mg^2+^ was present.

A representative trace is demonstrated in Figure 2A. After adding 30mM Mg^2+^, 39E stayed in the folded state (0.48) for a while, then turned into the extended state (0.12) (Figure 2A,B), similar to what happened with UO_2_^2+^. We verified the presence of acceptor molecules through the direct excitation of Cy5 using a 640 nm laser at the end of recordings to ensure this decline of E_FRET_ was not the byproduct of Cy5 photobleaching. Moreover, an even higher FRET (0.58) state was observed when Mg^2+^ was in solution (Figure 2C,D), indicating a more compact conformational state (C) of 39E. In addition, to collect more data, we recorded movies for a more extended period of time after injecting Mg^2+^. Traces showing the dynamics of 39E switching between different conformational states were captured (Figure 3). Not only can 39E switch between F and C states (Figure 3A) but also between F and E states (Figure 3B). We categorized the events that occurred during these switches into six possibilities: (i) stay at F then switch to C; (ii) stay at C then switch to F; (iii) stay at F then switch to E; (iv) stay at U then switch to F; (v) stay at C then switch to E and (vi) stay at E then switch to C. Using a hidden Markov model, we derived the dwell time of 39E staying at each state. Since only 18 traces showed the transition from C state to E state, and 3 traces showed the transition from E state to C state, these two data sets were excluded in the analysis. The rate of each state transition was obtained with a single exponential fitting: k_F–C_ = 0.67 s^−1^, k_C–F_ = 7.24 s^−1^, k_F–E_ = 0.015 s^−1^, and k_E–F_ = 0.19 s^−1^ (Figure 3C,D). It follows that 39E is more likely to stay in the folded state (F) when Mg^2+^ ions are present. Occasionally, it can switch to a more compact state (C) in which stem I and stem II are closer, or to the extended state like in the condition interacting with UO_2_^2+^. The observed FRET transitions are compiled into a transition density plot (TDP, Figure 3E) [43], revealing two distinct species of F↔C and F↔E transitions. Though 39E switched to the compact state more often, the life time of the C state is much shorter than the E state. In order to clearly show the transition probability, we also made a transition occupancy density plot (TODP) scaled by the fraction of molecules in F↔C transition (Figure 3F) [44]. It shows a roughly equal possibility of transition F↔C (59%, 336 molecules) and F↔E (52%, 296 molecules) with Mg^2+^ in solution.

To evaluate the effect of Mg^2+^ ions on the DNAzyme conformational dynamics, we also conducted experiments introducing different concentrations of Mg^2+^. Dwell time histograms of these states at each concentration are arranged in Appendix A. The lifetimes of individual states are closely comparable under diverse conditions (Appendix A). According to bulk experiments, the binding affinity of Mg^2+^ to 39E is approximately 5 mM [26,27]. The resemblance of behaviors of 39E switching between different states at 10 mM and 100 mM Mg^2+^ conditions imply that those state transitions may not be the outcome of the binding and dissociation of Mg^2+^ ions to the DNAzyme.

### 2.3. Conformational Changes without Divalent Metal Ions

Since varied Mg^2+^ concentrations suggested no particular tendency towards conformational dynamics of 39E, it is more likely that 39E has an intrinsic conformational frustration due to the local free energy minimum. To test this hypothesis, we recorded the movies under the conditions without any divalent metal ions in the imaging buffers. Overall, about 34% of the traces (n = 632 out of 1875) exhibited E_FRET_ changes between the native state (N) and the compact state (C) or the native state (N) and the extended state (E) (Figure 4A,B). However, neither C–E transitions nor other transitions towards the folded state (F) were observed without Mg^2+^. Fitting the dwell time histograms, we acquired the switching rates for individual transitions: native–to–compact state transition, k_N–C_ = 0.36 s^−1^ and k_C–N_ = 2.74 s^−1^; native–to–extended state transition, k_N–E_ = 0.33 s^−1^ and k_E–N_ = 0.58 s^−1^, respectively (Figure 4C,D). Unlike the condition with Mg^2+^ presence, 39E shows a favorable transition towards N↔C (87%, 549 molecules) compared to N↔E (19%, 122 molecules) transitions (Figure 4E,F). Though 39E can spontaneously switch to the E state similar to when UO_2_^2+^ is present, it shows no cleavage activity without divalent metal ions from bulk experiments, implying this kind of structural configuration might be the co–condition for DNAzyme’s function. Compared to the transition rates when Mg^2+^ ions were present, the stable conformational state of 39E shifted from the native state to the folded state. In the meanwhile, the transition rate between the stable state and the extended state (k_N–E_ = 0.33 s^−1^ for N to E transition and k_F–E_ = 0.015 s^−1^ for F to E transition with 30mM Mg^2+^) decreased dramatically by the binding of Mg^2+^, indicating 39E has been stabilized in the folded state (F), which would prevent the substrate strand from being cleaved.

### 2.4. Free Energy Landscapes of 39E Conformation Dynamics with and without Divalent Metal Ions

Based on the donor–acceptor distances calculated from the average FRET efficiency for each state of 39E, we proposed the secondary configurations of 39E stem I–II as shown in Appendix A. With the transition rate analysis under conditions with or without divalent metal ions, we were able to draw the energy landscape of 39E along the reaction path when introducing different metal ions (Figure 5). Without any divalent metal ions, 39E is stable at the native state (N). However, it can still transform into a compact state (C) or an extended state (E). A lower energy barrier of the C–N transition than that of the E–N transition marks 39E’s instability in the C state. After introducing Mg^2+^ in the system, 39E can swiftly switch to the folded state (F). Then, the barrier from the F state to the E state was dramatically raised (the quantitative energy of each transition is listed in Appendix A). This elevation might have happened by binding Mg^2+^ to the bulge loop region of the active site, which prevents cleavage activity. With the presence of UO_2_^2+^, 39E cannot form a compact formation because of the relatively larger ionic radii of UO_2_^2+^. Under such conditions, 39E tends to be stable in the extended state. Overall, 39E undergoes conformational switches with or without the divalent metal ions. It might facilitate 39E cleavage activity by the prearrangement of the extended configuration with the presence of UO_2_^2+^ and protect the substrate from being cleaved by trapping the 39E into the folded or compact configuration when Mg^2+^ is present.

## 3. Materials and Methods

### 3.1. Chemicals

The sequence of the DNAzyme 39E used in this study was adapted from [25], with some minor revisions. DNA was synthesized by GenScript, Nanjing, China. In particular, the substrate oligos (3′–GTGCAGGTAGAGAAGGATATCACTCA–5′) were labeled with Cy3 at the 5′ end, and the enzyme oligos (3′–TTTTTGTTGAGTGATACAGACTTCCAGCCAAATTGATGGGCTGACGTCTCTACCTGCAC–5′) were labeled with Cy5 in the middle and biotin at the 3′ end. The annealing process was conducted on a PCR machine heated up to 85 °C, then slowly cooled down to 20 °C over 4 h. Products of a final concentration of 2 μM were used without further purification.

Glucose, glucose oxidase, catalase and Trolox (6–hydroxy–2,5,7,8–tetramethylchromane–2–carboxylic acid) were commercially available from Sigma-Aldrich, St. Louis, MO, USA. Streptavidin was a product of Pierce, Thermo Fisher, Waltham, MA, USA. Other chemicals used in the experiments were of reagent grade. All buffers were filtered with 0.22 μm filters before use.

### 3.2. smFRET Experiment

Single–molecule measurements were carried out on an Olympus IX–71–based objective–type total internal reflection fluorescence (TIRF) microscope equipped with an oil immersion UAPON 100XOTIRF objective lens (N.A. = 1.49, Olympus).

All experiments were performed in an imaging buffer containing 50 mM MES (pH~6.5) and 50 mM NaCl with an oxygen scavenging system (0.8% w/v glucose, 1mg/mL glucose oxidase, 0.04 mg/mL catalase and 2 mM Trolox). The annealed DNAzyme sample was immobilized on a passivated glass surface through biotin–streptavidin interaction as previously described [45]. A 30 μL imaging buffer containing certain concentration of MgCl_2_ or UO_2_(OAc)_2_ (uranyl acetate) was injected into the flow cell to initiate the reaction at a particular time (between 10 s and 40 s after the recording started). For the experiments without divalent metal ions, EDTA was added in the imaging buffer to a final concentration of 0.5 mM as a chelator. Cy3 and Cy5 were excited by a 532 nm and a 640 nm laser. To exclude the possibility that the vanishment of the acceptor signals was due to Cy5 photobleaching, a direct excitation of Cy5 using a 640 nm laser was briefly executed (~2 s) within or at the end of the recording windows. The emissions were split into two channels by a dichroic mirror (FF640–FDi01, Semrock, St. West Henrietta, NY, USA) with two band–pass filters for the donor channel (FF01–580/40, Semrock) and the acceptor channel (FF01–675/67, Semrock), respectively. Signals were collected by an EMCCD (iXon 897, Andor Technology, Belfast, UK) with a temporal resolution of 50 ms. The FRET efficiency was calculated by the equation E_FRET_ = I_A_/(I_A_ + I_D_) [46].

### 3.3. Free Energy Estimation

Free energy calculation is based on Arrhenius equation k = A·exp(-E_a_/RT), where k is the reaction rate, A is the pre–exponential factor; E_a_ is the activation energy; R = 8.31 J/K·mol is the gas constant, and T = 297 K is the room temperature in Kelvin. Therefore, the energy of each transition can be calculated as E_a_ = RT·(lnA-lnk) (J/mol). The pre–exponential factor A is estimated as 10^3^ s^−1^ in our case, and the rate constant k of each transition is derived from smFRET data.

## 4. Conclusions

In summary, we used the smFRET technique to investigate the metal–ion–dependent folding and extending behaviors of DNAzyme 39E. In contrast to bulk experiments, a higher FRET state was observed, suggesting that 39E can form a more compacted configuration than the folded state. More interestingly, 39E can switch to the extended and compact state spontaneously in the absence of any divalent metal ions, indicating an intrinsic conformational dynamics of 39E DNAzyme. Compared to another smFRET study of 8–17 DNAzyme, 39E undergoes an extending configuration even with an uncleavable substrate strand. Though Mg^2+^ or Zn^2+^ induces a global folding of 8–17, the 8–17 DNAzyme still needs to go through an unfolding path to activate its cleavage function. Different from this mechanism, the folded state of 39E stabilized by Mg^2+^ might be a protective state that prevents the substrate strand from being cleaved by the enzyme strand. The transitions between the stable states (N or F state) and the compact state or the extended state imply that the catalytic function of 39E is not simply triggered only by the allosteric arrangement but also by the interaction with UO_2_^2+^ ions. Future work will be performed to test the effects of different mutations in the catalytic core on the conformational dynamics of 39E. In order to obtain a deeper insight into the correlations of the structure and function of 39E DNAzyme, the three–color FRET method will be needed to investigate the configurations of three stems at the same time.

## Figures and Tables

**Figure 1 ijms-24-01212-f001:**
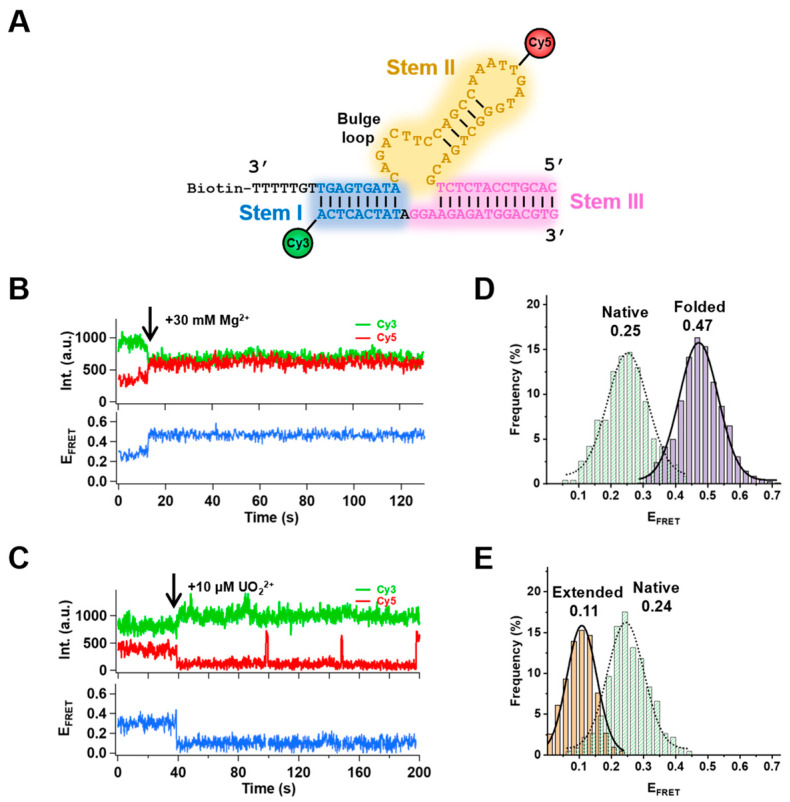
39E DNAzyme structure and the conformational changes with the injections of Mg^2+^ or UO_2_^2+^ in solution. (**A**) The secondary structure of 39E DNAzyme. The substrate strand was modified with the fluorescent dye Cy3 at the 5′ end, and the original cleavage site rA was replaced with a deoxynucleotide adenine (black “A”). The enzyme strand was labeled with biotin at the 3′ end and a fluorescent dye group Cy5 on the distal loop. (**B**) Time traces of Cy3 and Cy5 signals and FRET changes. The arrow denotes the addition of 30 mM Mg^2+^ at 13 s. (**C**) Fluorophore signals and E_FRET_ time trace with the addition of 10 μM UO_2_^2+^ at 38 s, the acceptor’s presence was checked by direct excitation of Cy5 fluorophore using a 640 nm laser. Histograms of different FRET states with Mg^2+^ (752 traces, E_FRET_~0.25 for native state and E_FRET_~0.47 for the folded state) (**D**) and with UO_2_^2+^ (286 traces, E_FRET_~0.24 for native state and E_FRET_~0.11 for the extended state) (**E**) in the solutions.

**Figure 2 ijms-24-01212-f002:**
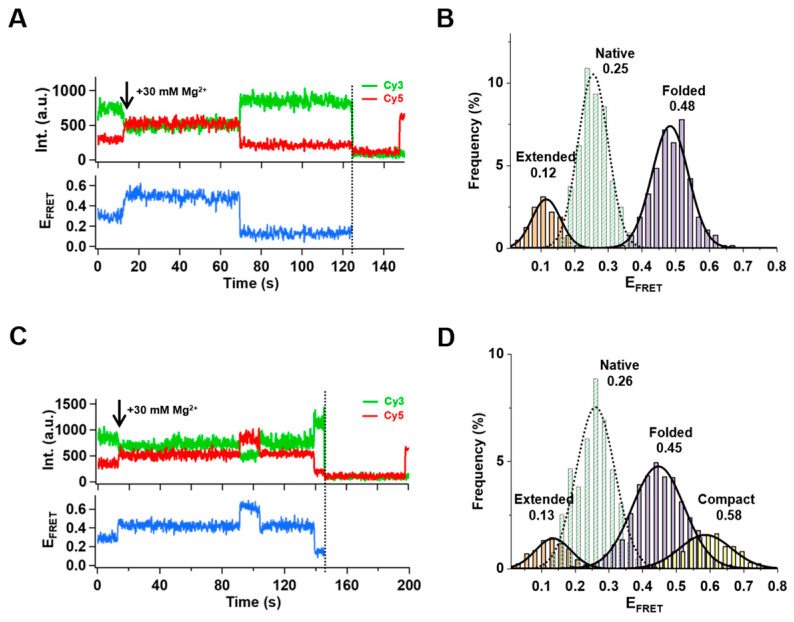
Mg^2+^–dependent conformational switches of 39E DNAzyme. (**A**) The time trace of fluorescence of donor and acceptor and sequential photobleaching of Cy3 (dashed line). (**B**) With the presence of 30 mM Mg^2+^, three different levels of FRET histograms from individual FRET time trajectories (312 traces) were obtained. E_FRET_~0.25 for the native state (N) before adding Mg^2+^ solution and E_FRET_~0.48 for the folded state (F) after injecting Mg^2+^, and E_FRET_~0.12 for the extended state (E). (**C**) In the time between the addition of Mg^2+^ and photobleaching of the donor fluorophore (dashed line), 39E DNAzymes undergo four conformational changes. (**D**) The histograms of four different FRET levels obtained from individual FRET traces (219 traces). E_FRET_~0.26 for native state (N) before adding Mg^2+^ solution and E_FRET_~0.45 for the folded state (F) after injecting Mg^2+^, E_FRET_~0.12 for extended state (E) and an additional E_FRET_~0.58 for a compact state (C).

**Figure 3 ijms-24-01212-f003:**
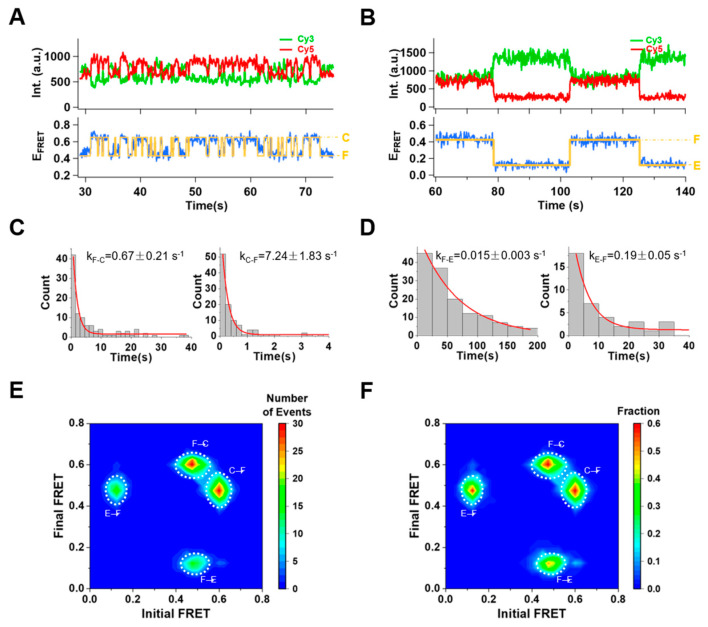
Conformation switches between different conformational states in presence of 30 mM Mg^2+^ in solution. (**A**) The time trace of fluorescence shows that the 39E DNAzymes switched between the folded (F) and the compact (C) state and (**B**) between the folded (F) and the extended (E) state. The orange lines present the hidden Markov model fittings. Dwell time histogram by single exponential fittings give the rate of each transition is k_F–C_ = 0.67 s^−1^, k_C–F_ = 7.24 s^−1^ (**C**), and k_F–E_ = 0.015 s^−1^, k_E–F_ = 0.19 s^−1^ (**D**). (**E**) The transition density plot (TDP) of F↔C (761 transitions) and F↔E (418 transitions) switches. (**F**) Transition occupancy density plot (TODP) of F↔C (336 molecules) and F↔E (296 molecules) transitions scaled by the fraction of molecules in F↔C transition.

**Figure 4 ijms-24-01212-f004:**
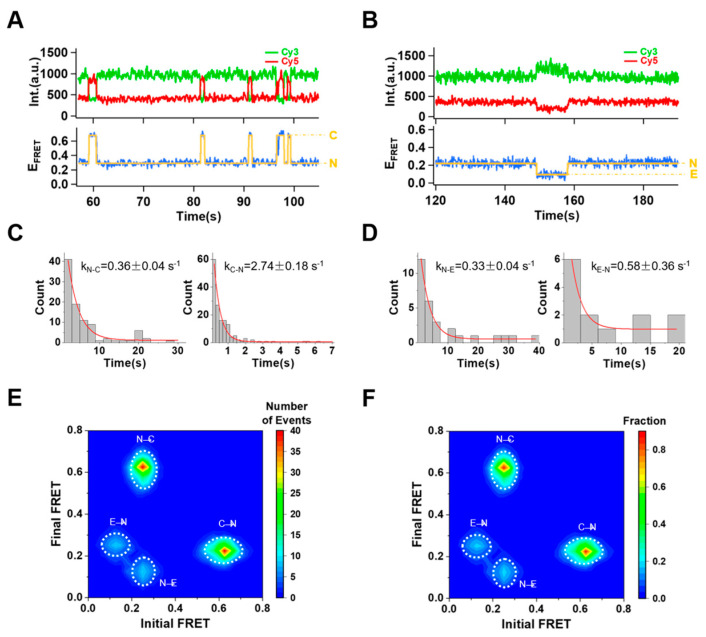
Intensity trajectories of 39E DNAzyme showing conformational dynamics without Mg^2+^. 39E DNAzymes switched back and forth between N state and C state (**A**) and between N state and E state (**B**). The transition rates derived from dwell time analysis are given in (**C**) and (**D**): k_N–C_ = 0.36 s^−1^, k_C–N_ = 2.74 s^−1^ and k_N–E_ = 0.33 s^−1^, k_E–N_ = 0.58 s^−1^. (**E**) The transition density plot (TDP) of the traces showing different transition populations of N↔C (1046 transitions) and N↔E (163 transitions). (**F**) Transition occupancy density plot (TODP) of N↔C (549 molecules) and N↔E (122 molecules) transitions scaled by the fraction of molecules in N↔C transition.

**Figure 5 ijms-24-01212-f005:**
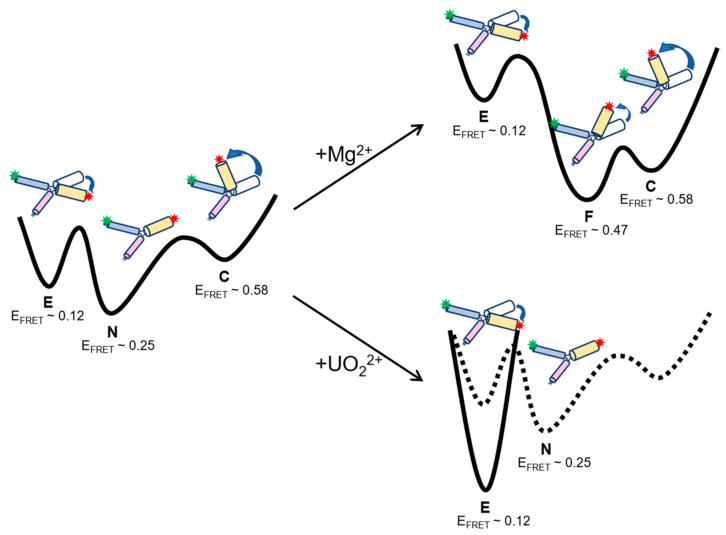
Free energy diagram based on the transition rates derived from smFRET experiments. E, N, F, and C represent the extended state, the native state, the folded state, and the compact state, respectively.

## Data Availability

All data are available in the main text or the Appendix A.

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
