# Peer review of "Single–Molecule Study of DNAzyme Reveals Its Intrinsic Conformational Dynamics"

_ijms, 2023, doi:10.3390/ijms24021212_

Round 1

Reviewer 1 Report

The current manuscript by Zhang et al. reports a smFRET study of a DNAzyme named 39E by identifying several conformational states. The authors expect that the present study will provide mechanistic insights into the 39E DNAzyme activity. There are several concerns and issues that may hamper the publication of this work. I would like to ask the authors to address these issues before any formal publication of this study, as described in the following.

(1)   Have the authors conducted smFRET experiments on functional 39E without any mutations? How are the conformational states identified on the mutated 39E mapped or related to those of the native or wildtype 39E? How does the mutation impact on the conformational selection of functional states and/or activities?

(2)   Is there an atomic model of 39E experimentally determined or previously published? If so, can the authors interpret their smFRET data in light of the 39E’s atomic model?

(3)   On page 6, the authors claim “the U state is the 39E function state”. Why is the native state not the functional state? What is the supporting evidence for this statement? What about the function of the C state? The discussion around the identities of the states and their corresponding functions are not clearly described and are not sufficiently justified with experimental data or supporting evidence.

(4)   The main conclusion of the paper is not well supported by the data due to lack of structural data and other aspects of supporting evidence. It is not convincing to make conclusion like the last sentence of the abstract.

(5)   The materials and methods section should include necessary details regarding the calculation of free energy profile exhibited in Figure 5.

Reviewer 2 Report

1.     The manuscript by Zhang, Y et al., titled “Single Molecule Study of DNAzyme Reveals Its Conformational Selection” studies metal-ion dependent conformational changes of the DNA enzyme (DNAzyme) 39E using single-molecule FRET (smFRET) spectroscopy. The uranyl-specific DNAzyme 39E catalyzes the cleavage of RNA. Similar to ribozymes, DNAzymes undergoes functionally important conformational changes which the authors investigated in this study. Using a non-cleavable mutant of the ribozyme, the authors first showed that 39E DNAzyme adopts four different conformations in the presence of Mg2+ ions – unfolded, native, folded and compact conformations. They show that the DNAzyme adopts predominantly unfolded conformation in the presence of UO22+ which is a cleavage competent state. This unfolded and compact state was also shown to be sampled in the absence of divalent metal ions, which led the authors to propose a conformational selection model involving a preorganized folded state of the DNAzyme. The authors propose a free energy diagram for the folding of the DNAzyme based on their smFRET data. Overall, while the manuscript is well written with clear figures and a nice model of the DNAzyme folding. However, the conclusion is short and does not discuss the importance and novelty of the study results, and does not compare and contrast the behavior of 39E in comparison to the other DNAzymes. Below are my comments to further improve the manuscript.

Major:

2.     In the introduction, the authors refer to stems I, II and II, while describing folding of the DNAzyme in the presence of different metal ions. However, it is hard to interpret these for readers not familiar with the system, without a description of the the secondary and/or tertiary structure of the DNAzyme in the text. Please describe briefly the (3-way junction) architecture in the introduction. Also given that the architecture for other DNAzymes is similar, and there are crystal structures available for some of them, it would be helpful to show the structure of a similar DNAzyme. This will help in correlating the FRET changes with structural changes, as shown in the final figure cartoon.

3.     Conclusion: Please describe the main findings of the study in relation to other similar studies on other DNAzymes. Why is the folding of 39E different from other ribozymes and represents a new model, as quoted on lines 79-80 in the introduction.

4.     The lowest FRET state is being referred to as an ‘unfolded’ (U) state, which I think is a misleading nomenclature. It is simply an extended state on the ribozyme, with different stacking configuration of the stems, resulting in the longest possible distance between the dyes, as shown in Fig. 5. This is also present in the 30 mM Mg2+ condition, supporting that it is not an ‘unfolded’ conformation. I suggest that the authors rename this as an Extended (E) state.

5.     Figure 1D and all FRET histograms: Do the FRET histograms shown here include the traces before the addition of metal ions? If yes, the histogram in the absence of any metal ion (0-10 s here) should be shown separately and not combined with portions after addition of metal ions. The initial portions of traces represent different condition with no metal ions and should be plotted separately to show equilibrium distribution of different conformations in that condition. The corrected histograms should represent the equilibrium population of two (or more) states which should correlate with their rate constants. This applies to all the histograms in other figures too. Also, I suggest showing the % of the two FRET states in the figure.

6.     Figure 1C: What are the spikes in the Cy5 intensity? If this is direct excitation to check for the presence of the acceptor, please mention it in the figure legend.

7.     Figure 3: From traces 3A and 3B, it is clear that there are at least two different kinds of traces with distinct kinetics.

8.     Figure S4 is an important result, and I suggest that it should be moved into the main text. Also, why are there no increase/decrease in the rate constants at different Mg2+ concentrations? It almost seems like the ribozyme is not sensing Mg2+ at all.

9.     Figure 4: Please show the FRET histogram in the absence of metal ions, showing the % of different FRET states in this condition. Also, the authors did not seem to use EDTA for this condition. For any measurements in the absence of divalent metal ions, it is a general practice to include a small (0.1 mM) amount of EDTA to chelate any trace amounts of divalent metal ion impurities that are generally present in many buffers and chemicals. If the authors did not use EDTA, I suggest them to repeat the experiment with  0.1 mM EDTA to see if the observed dynamics can be reproducible.

10.  TODP: I strongly suggest showing a transition density plot (TDP) for each condition. This is a 2D plot that shows the qualitative and quantitative description of the different transitions (FRET changes) observed in all the molecules, and is routinely used in smFRET studies of nucleic acids. This is especially important when systems have more than 2 conformations, like the 39E DNAzyme has at least 3 different states. Please refer to Blanco et al., Methods Enzymol. 2010; 472: 153–178 for TODPs)

11.  Lines 177 and 208: The authors loosely refer to ‘conformational selection’ of 39E at multiple places in the manuscript. However, to ascertain that the folding or metal-ion binding proceeds through conformational selection requires extensive kinetic analyses at different ligand (metal ion) concentrations. The authors should remove this or describe more clearly why they interpret their data to conform to the conformational selection pathway.

Minor

1.     Line 70: Correct a donor (Cy3) at the 5’ end of stem III to 5’ end of stem I

2.     Figure 1 and all histograms: Show the number of molecules (N = ?) analyzed for each histogram

3.     Line 204: correct ‘activate’ site to ‘active’ site

4.     Line 236: chemical formula for uranyl acetate is not UO2 (2+) and needs to be corrected

5.     Line 246: Delete the period ‘.’ In front of Conclusions

Round 2

Reviewer 1 Report

The authors have addressed all my questions adequately. I would like to recommend its publication.

Author Response

Thanks to the reviewer for the efforts to improve the quality of the manuscript. We really appreciate it.

Reviewer 2 Report

The authors have responded satisfactorily to most of my previous comments on the original manuscript titled “Single Molecule Study of DNAzyme Reveals Its Conformational Selection”. The revised version of the manuscript looks much better. The inclusion of TDP/TODPs and the revised conclusion look nice. There some minor changes the authors need to make in the manuscript, that are listed below.

Minor

1.     Using UO2 2+ (uranyl dioxide) as a shortened version of UO2(OAc)2 is not correct. Change UO2 to UO2 (OAc)2 everywhere in the manuscript or use UAc for abbreviation, after defining it first.

2.     Line 157: Change ‘extending’ to ‘extension’

Author Response

  1. Using UO2 2+ (uranyl dioxide) as a shortened version of UO2(OAc)2 is not correct. Change UO2 to UO2 (OAc)2 everywhere in the manuscript or use UAc for abbreviation, after defining it first.

We apologize for this misunderstanding. What we meant was that the source of UO22+ used in the experiment was uranyl acetate solution, like Mg2+ was from MgCl2 solution. Since 39E was selected with specificity towards uranyl ions (UO22+), we used the same chemical compound as in the reference 25 (Liu et al. 2007.PNAS. A Catalytic Beacon Sensor for Uranium with Parts-Per-Trillion Sensitivity and Millionfold Selectivity). To avoid any confusions, we revised the description as “30 μL imaging buffer containing certain concentration of MgCl2 or UO2(OAc)2 (uranyl acetate) was injected into the flow cell to initiate the reaction” in line 277-278.

  1. Line 157: Change ‘extending’ to ‘extension’

Corrected. Now in line 133.